# Investigation of the Nanoparticulation Method and Cell-Killing Effect following the Mitochondrial Delivery of Hydrophobic Porphyrin-Based Photosensitizers

**DOI:** 10.3390/ijms25084294

**Published:** 2024-04-12

**Authors:** Rina Naganawa, Hanjun Zhao, Yuta Takano, Masatoshi Maeki, Manabu Tokeshi, Hideyoshi Harashima, Yuma Yamada

**Affiliations:** 1Faculty of Pharmaceutical Sciences, Hokkaido University, Sapporo 060-0812, Japan; 2Graduate School of Environmental Science, Hokkaido University, Sapporo 060-0810, Japantak@es.hokudai.ac.jp (Y.T.); 3Research Institute for Electronic Science, Hokkaido University, Sapporo 010-0020, Japan; 4Graduate School of Engineering, Hokkaido University, Sapporo 060-8628, Japan; 5Fusion Oriented Research for Disruptive Science and Technology (FOREST) Program, Japan Science and Technology Agency (JST), Saitama 332-0012, Japan

**Keywords:** mitochondria, photodynamic therapy, mitochondrial delivery, MITO-Porter, photosensitizer

## Abstract

Photodynamic therapy is expected to be a less invasive treatment, and strategies for targeting mitochondria, the main sources of singlet oxygen, are attracting attention to increase the efficacy of photodynamic therapy and reduce its side effects. To date, we have succeeded in encapsulating the photosensitizer rTPA into MITO-Porter (MP), a mitochondria-targeted Drug Delivery System (DDS), aimed at mitochondrial delivery of the photosensitizer while maintaining its activity. In this study, we report the results of our studies to alleviate rTPA aggregation in an effort to improve drug efficacy and assess the usefulness of modifying the rTPA side chain to improve the mitochondrial retention of MITO-Porter, which exhibits high therapeutic efficacy. Conventional rTPA with anionic side chains and two rTPA analogs with side chains that were converted to neutral or cationic side chains were encapsulated into MITO-Porter. Low-MP (MITO-Porter with Low Drug/Lipid) exhibited high drug efficacy for all three types of rTPA, and in Low-MP, charged rTPA-encapsulated MP exhibited high drug efficacy. The cellular uptake and mitochondrial translocation capacities were similar for all particles, suggesting that differences in aggregation rates during the incorporation of rTPA into MITO-Porter resulted in differences in drug efficacy.

## 1. Introduction

Photodynamic therapy is a less invasive therapy that selectively kills cells at the site of irradiation [1,2]. In this therapy, a photosensitizer converts intracellular oxygen to singlet oxygen (^1^O_2_) in response to light, and then ^1^O_2_ induces death in surrounding cells [3,4]. ^1^O_2_ has a short lifespan and a risk of causing photosensitivity at sites other than the target site [5,6]. Therefore, the target-selective delivery of photosensitizers is necessary to increase their efficacy and reduce their side effects. As a promising approach, cancer phototherapy targeting mitochondria, the main organelle that produces ^1^O_2_, has been the subject of various studies [7,8,9].

The majority of reported mitochondrial delivery mechanisms involve the modification of photosensitizers with mitochondria-transferable elements. However, this strategy is only applicable to small molecules, and it reduces the activity of the photosensitizers through chemical bonding [10]. Encapsulating photosensitizers into nanocarriers is expected to be an effective approach to solve these problems. To date, we successfully developed MITO-Porter as a mitochondria-targeted nano carrier. MITO-Porter consists of mitochondria-fusogenic lipids and octaarginine (R8) with cellular uptake and mitochondrial targeting activities. Through fusion with the mitochondrial membrane, MITO-Porter enables the delivery of various molecules, ranging from small to large, into the mitochondria. Our laboratory has succeeded in encapsulating the photosensitizer rTPA into the MITO-Porter, and established a system to evaluate its anti-tumor effects. The anti-tumor effects confirmed in HeLa cells were derived from human cervical cancer and SAS cells derived from human tongue cancer, and anti-tumor effects were also observed in the evaluation using tumor-bearing mice. As the absorption wavelength of rTPA is higher than those of currently approved photosensitizers, it is also expected to exhibit improved light absorption efficiency in vivo.

Porphyrin-based molecules are one of the most studied photosensitizers. It is difficult to encapsulate them in nanocarriers because of their high aggregation properties, and these molecules are deactivated when aggregated within the particles [11]. rTPA, an amphiphilic porphyrin-based photosensitizer, has been successfully incorporated into MITO-Porter [11]. We previously reported that rTPA encapsulated MITO-Porter was prepared using two different ratios of rTPA to lipids, and the singlet oxygen production ability was compared. As a result, the rTPA encapsulated MITO-Porter with a lower ratio of rTPA to lipids (Low-MP) exhibited a higher singlet oxygen production ability compared to the rTPA encapsulated MITO-Porter with a higher ratio of rTPA to lipids (High-MP). Observation of the internal structure of particles using electron microscopy revealed a multilamellar structure in Low-MP, while a granular structure was observed in High-MP. Thus, it was suggested that dispersing rTPA between the lipid bilayers of particles without aggregation is possible, and dispersing it between the lipid bilayers results in a higher singlet oxygen production ability [11].

One feasible explanation for the good dispersibility of rTPA is that the negative charge of its side chains reduces its aggregation properties, but the effect of the charge on the dispersion of rTPA and drug efficacy has not been clarified. However, the negative charge of the side chains of rTPA might reduce its retention in negatively charged mitochondria. Therefore, to develop rTPA@MITO-Porter, which will be highly functional after mitochondrial transfer, it is necessary to verify the necessity of alleviating rTPA aggregation to improve its efficacy and the usefulness of modifying the side chains of rTPA to improve its mitochondrial retention.

Towards this aim, we prepared MITO-Porter containing two types of rTPA derivatives in which the side chains were converted to neutral or cationic. We compared the efficacy of MITO-Porter containing three types of rTPAs (negative-rTPA, positive-rTPA, and neutral-rTPA; Figure 1). Cationic rTPA-encapsulated MITO-Porter was prepared with the expectation of improved retention in negatively charged mitochondria (Figure 2). Following the measurement of the physical properties of the prepared particles, the in vitro capacity for ^1^O_2_ production was assessed utilizing the Singlet Oxygen Sensor Green (SOSG) assay, and the cell-killing effect on HeLa cells was evaluated by the WST-1 assay. The particle kinetics was evaluated by flow cytometry for intracellular uptake and by microscopic observation for mitochondrial translocation.

## 2. Results

### 2.1. Construction of MITO-Porter Encapsulating Different rTPA Derivatives

As with negative-rTPA@MITO-Porter, we attempted to prepare two variations of rTPA@MITO-Porter derivatives employing a microfluidic device. With particle preparation using microfluidic devices, we can control the particle size by changing the flow rate ratio between the lipid and aqueous phases. The preparation conditions for the negative-rTPA@MITO-Porter were similar to those for the MITO-Porter, where a mitochondria-fusogenic lipid suspension with a lipid composition of 1,2-Dioleoyl-sn-glycero-3-phosphatidylethanolamine (DOPE)/cholesterol/1,2-Distearoyl-sn-glycerol-3-phosphoethanolamine-N-[amino(polyethylene glycol)-2000] (DSPE-PEG2000)/rTPA (=9:2:0.22:X, molar ratio) was prepared. Particles were prepared by mixing and stirring the organic phase and the aqueous phase using a microfluidic device under the flow rate conditions of organic phase: 150 (µL/min), aqueous phase: 350 (µL/min), and total flow rate: 500 (µL/min) (Figure 3).

We prepared MITO-Porter with two rTPA encapsulation rates and Drug/Lipid (D/L) ratios of 0.5 (Low-MP) and 2.5 mol% (High-MP). Consequently, we succeeded in preparing particles with a size of 60–80 nm and a PDI (polydispersity index) of 0.150–0.230, regardless of the charge and rTPA encapsulation rate (Table 1). The rTPA encapsulation rate was 70–80% for all particles, indicating that rTPA was successfully encapsulated, even in High-MP, where rTPA aggregation is likely to occur.

### 2.2. Evaluation of the ^1^O_2_ Production Abilities of the Three Types of rTPA-Encapsulated MITO-Porter Constructs

We compared the ability of the three rTPA@MITO-Porter constructs to produce ^1^O_2_ in vitro. In this experiment, the amount of singlet oxygen produced was quantified by adding the singlet oxygen-sensitive probe SOSG to a solution of rTPA@ MITO-Porter, followed by 12.3 J/cm^2^ light irradiation (60 s) and calculation of the fluorescence intensity change before and after irradiation. Fluorescence was excited in proportion to singlet oxygen by the SOSG probe. To compare the singlet oxygen production capabilities of three types of rTPA@MITO-Porter in test tubes, the singlet oxygen production ability was evaluated in vitro. When comparing the results among different amounts of rTPA encapsulation, the singlet oxygen production ability of Low-MP was significantly higher than that of High-MP for all three types of rTPA (Figure 4 and Appendix A). Furthermore, when comparing the results among the three types of rTPA, the charged rTPA@MITO-Porter exhibited significantly higher singlet oxygen production ability than the uncharged MITO-Porter.

Based on the observation of particle structures of negative-rTPA@MITO-Porter in previous studies, it was noted that the high singlet oxygen production ability of Low-MP with a layered structure and the low singlet oxygen production ability of High-MP with a granular structure, suggesting aggregation of rTPA in High-MP [11]. Therefore, the difference in singlet oxygen production ability of particles due to rTPA species and rTPA encapsulation amount may be attributed to differences in the degree of rTPA aggregation within the particles. Hence, further investigation is needed to understand the possibility of aggregation occurring in High-MP for all three types of rTPA, and the potential contribution of the charge of rTPA side chains to the alleviation of rTPA aggregation.

### 2.3. Evaluation of the Cell-Killing Effects of the rTPA-Encapsulated MITO-Porter Constructs

To verify whether the three rTPA@MITO-Porter constructs exhibit identical drug effects in cellular systems as observed in cell-free systems, we evaluated the phototoxicity of the three rTPA@MITO-Porters in HeLa cells. In this experiment, three types of rTPA@MITO-Porter were added to HeLa cells, followed by 12.3 J/cm^2^ light irradiation (60 s) after 3 h, and the cell viability 2 h after irradiation was determined using the WST-1 Assay. Comparing the three Low-MP types, negative-rTPA exhibited the strongest cell-killing effect, followed by positive-rTPA and neutral-rTPA (Figure 5A). Comparing the three High-MP types, neutral-rTPA displayed the strongest cell-killing effect, followed by positive-rTPA and negative-rTPA (Figure 5B). Comparing the results among the different rTPA encapsulation rates, Low-MP had a significantly higher cell-killing effect than High-MP for all three rTPA types.

These results are consistent with those for ^1^O_2_ production in the cell-free system, suggesting that no difference in mitochondrial retention occurred among the three rTPA@MITO-Porter constructs and that the constructs function similarly in different in vitro systems. The comparison results of the intracellular observation of the three types of rTPA-MITO-Porter will be discussed in the following section.

### 2.4. Evaluation of the Intracellular Uptake of Three rTPA-Encapsulated MITO-Porter Constructs

The cellular uptake of the three constructs was evaluated by flow cytometry. In this experiment, three types of rTPA@MITO-Porter labeled with the fluorescent dye conjugated lipid, DOPE-N-(7-nitro-2-1,3-benzoxadiazole-4-yl) (NBD-DOPE), were added to HeLa cells for 1 h, and the fluorescence intensity of the particles taken up by the HeLa cells was detected using flow cytometry to evaluate the cellular uptake ability of the three types of rTPA@MITO-Porter. All three types of rTPA-MITO-Porter consist of mitochondria-fusogenic lipids with R8, which has cellular uptake and mitochondrial targeting activities, with only the charge of the encapsulated rTPA side chains differing. Since there were no differences in the physical properties of the particles among the different rTPA species, it is considered that all three types of rTPA@MITO-Porter are similarly taken up into the cells due to the action of R8.

According to the flow cytometry results, all three types of Low-MP were confirmed to have similar cellular uptake abilities to Empty-MITO-Porter (Figure 6). Similarly, all three types of High-MP were confirmed to have similar cellular uptake abilities to Empty-MITO-Porter (Appendix A). Based on these results, it is suggested that, regardless of the charge of the rTPA side chains, all three types of rTPA@MITO-Porter are taken up into the cells to a similar extent as Empty-MITO-Porter, regardless of the encapsulation amount (Low or High).

### 2.5. Evaluation of the Mitochondrial Translocation of the rTPA-Encapsulated MITO-Porter Constructs

To compare the mitochondrial translocation of the three rTPA@MITO-Porter constructs, their subcellular localization was evaluated by confocal laser scanning microscopy (CLSM). MITO-Porter labeled with 7-nitrobenz-2-oxa-1,3-diazone (NBD)-DOPE lipids appears green, mitochondria stained with MitoTracker Deep Red appear red, and colocalization is indicated in yellow. It is believed that all three types of rTPA@MITO-Porter accumulated in the mitochondria to a similar extent through the action of mitochondria-fusogenic lipids with R8, and that all three types of rTPA are delivered to the mitochondria via membrane fusion. The results of the observation showed colocalization with mitochondria for all three types of rTPA-MITO-Porter (Low, High), suggesting that both Low-MP and High-MP accumulate in the mitochondria (Figure 7 and Appendix A). Further investigation is needed to determine the difference in mitochondrial retention depending on the rTPA species.

## 3. Discussion

The singlet oxygen production ability and cytotoxic effects of the three types of rTPA-MITO-Porter were higher for Low-MP compared to High-MP across all three rTPA species, with the order of effectiveness being negative rTPA, positive rTPA, and neutral rTPA when comparing the three within Low-MP. Flow cytometry analysis and microscopic observations suggested that the cellular uptake ability and the mitochondrial co-localization of the three rTPA@MITO-Porter constructs were equivalent regardless of the charge and encapsulation rate of rTPA. Conversely, the ability of the three rTPA@MITO-Porter constructs to induce ^1^O_2_ production and kill cells differed depending on the charge of rTPA and the level of rTPA encapsulation. These results suggest that the charge and encapsulation of rTPA changed the degree of rTPA aggregation in rTPA@MITO-Porter, resulting in differences in drug efficacy.

Although these results suggest that negative rTPA is most suitable from the viewpoint of drug efficacy, and the three types of rTPA are equivalent from the viewpoint of mitochondrial transfer, the drug efficacy and mitochondrial retention could change as the addition time of MITO-Porter is varied. Specifically, it is possible that increasing the addition time will increase the retention of positive rTPA@MITO-Porter or, conversely, it could become more susceptible to degradation. In the future, we will conduct similar experiments by varying the addition time to deepen our consideration of the usefulness of positive-rTPA@MITO-Porter.

Based on the observation of particle structures of negative-rTPA@MITO-Porter in a previous study [11], it was noted that the high singlet oxygen production ability of Low-MP with a layered structure and the low singlet oxygen production ability of High-MP with a granular structure, suggesting aggregation of rTPA in High-MP. Therefore, the difference in singlet oxygen production ability of particles due to the rTPA species may be attributed to the potential alleviation of rTPA aggregation by the charged side chains of rTPA, and the difference in encapsulation amount may be due to rTPA dispersion within the lipid membrane in Low-MP compared to the inability of some rTPA to disperse within the lipid membrane in High-MP, leading to aggregation. If aggregation results in a decrease in singlet oxygen production ability, improving the dispersibility of rTPA within MITO-Porter is necessary for enhancing its pharmacological efficacy. Therefore, it is necessary to explore preparation conditions that improve the dispersibility of rTPA through verification of the presence or absence of aggregation by observing the internal structure of three types of rTPA@MITO-Porter and through the investigation of new lipid compositions.

It is also expected that the use of salt in cationic rTPA, as well as anionic rTPA, may further alleviate the aggregation of rTPA. Therefore, we would like to study this issue in the future.

## 4. Materials and Methods

### 4.1. Materials

The materials used in this study were as follows: Avanti Polar Lipids, Inc. (Alabaster, AL, USA) supplied cholesterol, DOPE and NBD-DOPE; NOF Corporation (Tokyo, Japan) supplied DSPE-PEG2000; Toray Research Center, Inc. (Tokyo, Japan) supplied stearylated R8 (STR-R8); Wako (Osaka, Japan) provided 4-(4,6-Dimethoxy-1,3,5-triazin-2-yl)-4-methylmorpholinium Chloride (DMT-MM) and Dulbecco’s Modified Eagle’s Medium (DMEM); Sigma-Aldrich Corp. (St. Louis, MO, USA) supplied fetal bovine serum (FBS); Takara Bio Inc. (Shiga, Japan) supplied the premix WST-1 Cell Proliferation Assay System kit and phosphate-buffered saline [PBS (−)] and; Thermo Fischer Scientific Inc. (Waltham, MA, USA) supplied SOSG reagent. HeLa cells were sourced from Riken BRC (Tsukuba, Japan). rTPA was synthesized and characterized according to established literature protocols. As microfluidic devices for this study, invasive Lipid Nanoparticle Production device (iLiNP) were used.

### 4.2. Preparation of rTPA and Its Derivatives

rTPA-NH_2_: rTPA (20.0 mg, 12.0 μmol) and DMT mm (20.0 mg, 60.0 μmol) were dissolved in 30 mL dry-THF and stirred for 1 h under argon atmosphere after 5 min of ultrasonication. Then, ethylenediamine (57.0 mg, 50.0 eq) was added to the reaction solution. After 4 h, the THF layer was washed with brine solution 3 times and dried over Na_2_SO_4_. The crude was further purified by reprecipitation from THF and hexane. The product was obtained as a black powder (15.0 mg, 75%). 1H NMR (DMSO-d6) 400 MHz δ = 9.71 (4H, d, J = 4.0 Hz), 8.66 (4H, d, J = 4.0 Hz), 8.33 (4H, brs), 8.21 (4H. brs), 7.79 (2H, d, J = 8.2 Hz), 7.18 (8H, d, J = 8.7 Hz), 7.01 (8H, d, J = 8.7 Hz), 6.53 (2H, s), 6.43 (2H, d, J = 8.2 Hz), 4.01 (4H, m), 3.80 (8H, d, J = 4.0 Hz), 3.79 (12H, s), 1.99 (4H, brs), 1.64 (4H, brs), 1.37 (4H, brs), 1.18 (4H, brs), 0.99 (20H, brs), 0.64 (6H, t, J = 6.6 Hz). HR-ESI-Mass: calcd. for C102H106N10O8Zn: 1664.7507 *m*/*z*, observed: 1664.7648 *m*/*z*. The 1H NMR spectra of rTPA-NH_2_ are shown in Appendix A.

rTPA-OH: rTPA (20.0 mg, 12.0 μmol) and DMT mm (20.0 mg, 60.0 μmol) were dissolved in 30 mL dry-THF and stirred for 1 h under argon atmosphere after 5 min of ultrasonication. Then, an excess amount of 2-aminoethanol (200 mg) was added to the reaction solution. After 4 h, the THF layer was washed with brine solution 3 times and dried over Na_2_SO_4_. The crude was further purified by reprecipitation from THF and hexane. The product was obtained as a black powder (15.5 mg, 76%). 1H NMR (DMSO-d6) 400 MHz δ = 9.71 (4H, d, J = 4.1 Hz), 8.66 (4H, d, J = 4.1 Hz), 8.31 (4H, d, J = 8.2 Hz), 8.20 (4H, d, J = 8.2 Hz), 7.79 (2H, d, J = 8.2 Hz), 7.19 (8H, d, J = 9.0 Hz), 7.00 (8H, d, J = 9.0 Hz), 6.51 (2H, s), 6.44 (2H, d, J = 8.2 Hz), 4.02 (4H, m), 3.81 (8H, br), 3.79 (12H, s), 1.99 (4H, brs), 1.61 (4H, brs), 1.36 (4H, brs), 1.13 (4H, brs), 0.99 (20H, brs), 0.60 (6H, t, J = 6.6 Hz). HR-ESI-Mass: calcd. for C102H104N8O10Zn: 1666.7167 *m*/*z*, observed: 1666.7120 *m*/*z*. The 1H NMR spectra of rTPA-OH are shown in Appendix A.

As shown in Appendix A, the 1H NMR spectra of rTPA-NH_2_ and rTPA-OH do not show prominent peaks due to impurities, and the purity of the compounds can be guaranteed to be greater than 90%, even including the common errors that arise from NMR measurements (typically ca. 5%).

### 4.3. Preparation of rTPA@MITO-Porter Using iLiNP

The iLiNP as microfluidic device (settings detailed in Figure 3) was employed for this study. An EtOH solution containing 1.3 mM lipids [DOPE:Chol:DSPE-PEG2000:rTPA (9:2:0.22:X, molar ratio)] and PBS (−) was prepared. We prepared rTPA-MITO-Porter with two different Drug/Lipid ratios (Drug/Lipid = 0.5 [mol%] for Low-MP, Drug/Lipid = 2.5 [mol%] for High-MP). For fluorescent labeling of rTPA@MITO-Porter, 0.5 mol% NBD-DOPE per total lipid was added to the EtOH solution. rTPA@MITO-Porter was prepared by mixing the lipids in EtOH and PBS (−) using the iLiNP, followed by dialysis against PBS (−). Physichochemical properties of prepared particles, including particle size, polydispersity index (PDI) and ζ-potential were measured, and the encapsulation efficiencies were calculated. The detailed procedures are described in the Appendix A.

### 4.4. Evaluation of In Vitro Photodynamic Therapy

To assess cell viability, cells were cultured on a 48-well plate (Corning Inc., Corning, NY, USA) for 24 h. Subsequently, cells cultured in DMEM (FBS−) were treated with the samples for 1 h, followed by incubation for 2 h in DMEM (FBS+). After washing with DMEM (FBS+), cells were exposed to photoirradiation using a xenon lamp at a wavelength of 700 ± 6 nm (68.3 mW/cm^2^). WST-1 reagent was added immediately post-irradiation, and the mixture was incubated for 2 h. Absorbance change of the reagent was measured at 450 nm with reference to 630 nm using a plate reader (Varioskan LUX, Thermo Fisher Scientific, Waltham, MA, USA).

### 4.5. Cellular Uptake Analysis

HeLa cells were seeded in six-well plates (Corning) at 24 h before the experiment and maintained in 5% CO_2_ at 37 °C. Following washing the cells with PBS (−), they were incubated with DMEM (FBS−), and the prepared particles labeled with NBD-DOPE was added for 1 h. The final concentration of total lipid of particle was 27.5 μM in the medium. After two washes with PBS (−) containing heparin, the cells were trypsinized and suspended in DMEM (FBS+). After centrifugation, the pellet was suspended in PBS (−) containing bovine serum albumin and sodium azide, followed by analysis via flow cytometry (CytoFLEX; Beckman Coulter Inc., Pasadena, MA, USA). The detailed procedures are described in the Appendix A.

### 4.6. Intracellular Observation Using CLSM

For the intracellular observation, the HeLa cells were seeded cells in 35 mm glass-bottom dishes (IWAKI, Tokyo, Japan) 24 h prior to the experiment, and maintained at 37 °C in 5% CO_2_. Following a wash with DMEM (FBS−), cells were exposed to DMEM (FBS−) containing rTPA@MITO-Porter labeled with NBD-DOPE. The final concentration of total lipid of particle was 27.5 μM in the medium. After 1 h incubation, the medium was replaced with fresh DMEM (FBS+), and cells were further incubated for 100 min. After this incubation period, cells were stained mitochondria red with MitoTracker Deep Red before observation. Following a wash with DMEM (FBS+), fresh DMEM (FBS+) was added, and CLSM images were captured using a Nikon-A1 microscope (NIKON CORPORATION, Tokyo, Japan). The detailed procedures are described in the Appendix A.

Further methodological details, including “rTPA@MITO-Porter using iLiNP”, “Photoinduced ^1^O_2_ generation detection”, ”Cellular uptake analysis”, “Intracellular observation by CLSM” and statistical analyses are included in the Appendix A.

## Figures and Tables

**Figure 1 ijms-25-04294-f001:**
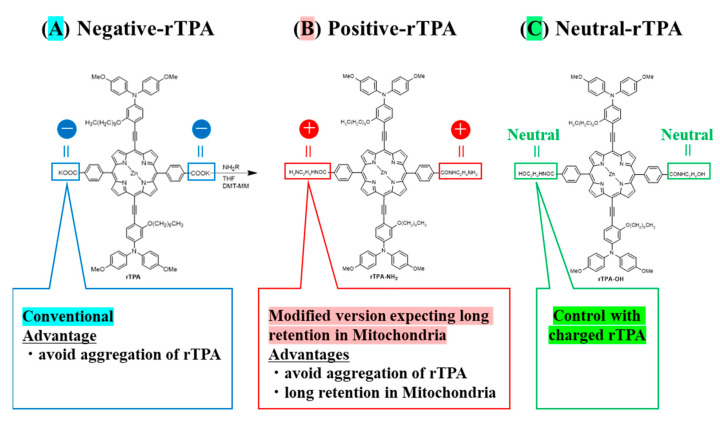
Structure of three types of rTPA ((**A**) rTPA, (**B**) rTPA-NH2, (**C**) rTPA-OH). The side chains of (**B**,**C**) are modified from (**A**).

**Figure 2 ijms-25-04294-f002:**
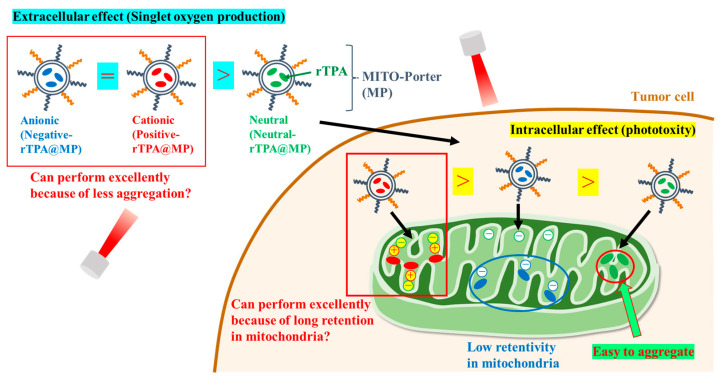
Schematic diagram showing the aim of the intracellular dynamics of the three rTPA-MITO-porters. MITO-Porter reaches mitochondria after cellular uptake.

**Figure 3 ijms-25-04294-f003:**
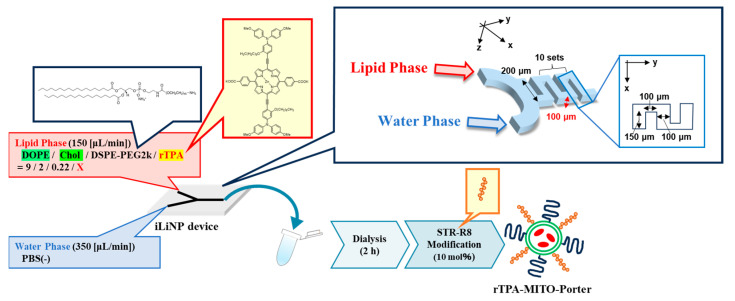
Schematic illustration depicting concept of preparing MITO-Porter. The preparation involves the utilization of a microfluidic device and a dialysis process. In the green circle of rTPA-MITO-Porter, DOPE and Cholesterol are represented, while blue and orange rods on the surface of rTPA-MITO-Porter represent DSPE-PEG2000 and stearylated R8 (STR-R8), respectively.

**Figure 4 ijms-25-04294-f004:**
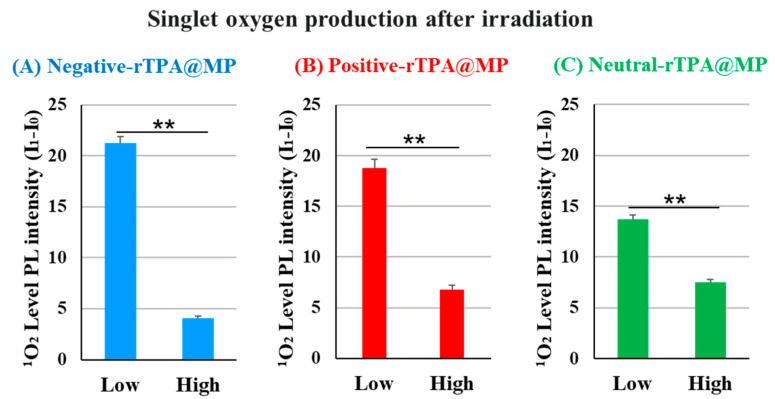
Validation of the singlet oxygen production ability of rTPA in MP ((**A**) negative-rTPA@MP, (**B**) positive-rTPA@MP, (**C**) neutral-rTPA@MP). Data are means ± S.D. (n = 3–4), ** *p* < 0.01 by unpaired *t*-test. Singlet oxygen quantification is expressed as the difference in photoluminescence (PL) intensity (I_1_-I_0_). It is PL intensity after 12.3 J/cm^2^ light irradiation of 60 s.

**Figure 5 ijms-25-04294-f005:**
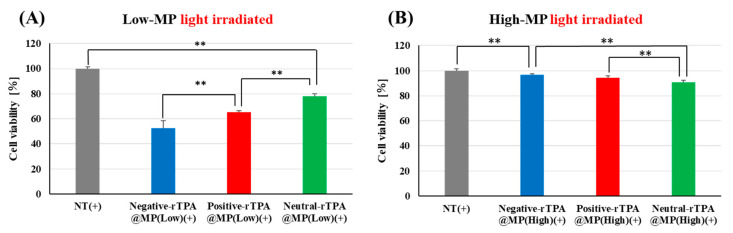
Validation of the phototoxicity of three-MITO-Porters by WST-1 assay with HeLa cell. (**A**) Comparison of cell viabilities between three types of Low-MP with light irradiated (12.3 J/cm^2^, 60 s). NT(+) means light-irradiated control without MITO-Porter. (**B**) Comparison of cell viabilities between three types of High-MP with light irradiated. The data are presented as means + SD (n = 3). Statistical significance was determined by non-repeated ANOVA, followed by SNK, with ** *p* < 0.01.

**Figure 6 ijms-25-04294-f006:**
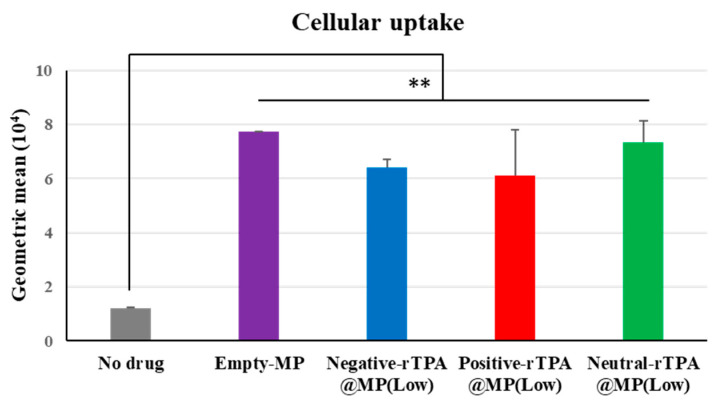
Flow cytometry analysis to evaluate the cellular uptake of Low-MP. Flow cytometry analysis was conducted to assess the cellular uptake of Low-MP, with cellular uptake expressed as the mean fluorescence intensity. The data are presented as means ± SD (n = 3). Statistical significance was determined by non-repeated ANOVA, followed by SNK, with ** *p* < 0.01.

**Figure 7 ijms-25-04294-f007:**
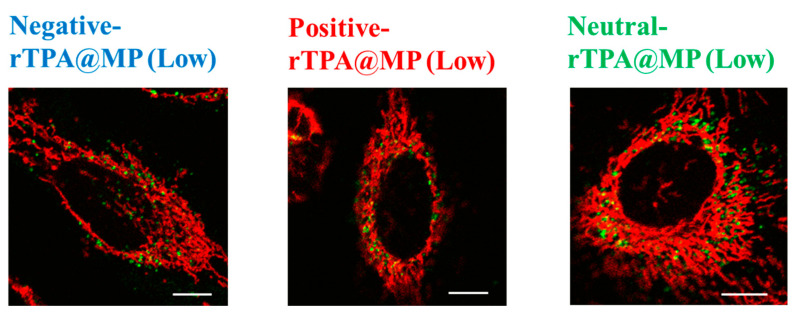
Intracellular observation of Low-MP using CLSM. Low-MPs (colored green) were observed to overlap with mitochondria stained in red within HeLa cells, resulting in yellow signals in the combined image. The scale bars represent 10 μm. HeLa cells were seeded cells in 35 mm glass-bottom dishes 24 h prior to the experiment.

**Table 1 ijms-25-04294-t001:** PhyPsicochemical properties of MITO-porter.

	Drug/Lipid (mol%) (Prepared)	Particle Size (nm)	PDI	ζ -Potential (mV)	Encapsulation Rate (%)	Drug/Lipid (mol%)
Negative-rTPA@MP	0.5 (Low)	62 ± 5.8	0.18 ± 0.052	18 ± 2.5	78 ± 6	0.46 ± 0.053
	2.5 (High)	67 ± 9.6	0.23 ± 0.078	17 ± 3.4	72 ± 8	2.28 ± 0.079
Positive-rTPA@MP	0.5 (Low)	66 ± 3.8	0.15 ± 0.052	19 ± 1.5	76 ± 3	0.47 ± 0.041
	2.5 (High)	74 ± 6.2	0.16 ± 0.067	18 ± 2.1	75 ± 6	2.21 ± 0.056
Neutral-rTPA@MP	0.5 (Low)	81 ± 11	0.22 ± 0.074	21 ± 2.0	71 ± 9	0.42 ± 0.059
	2.5 (High)	81 ± 8.8	0.17 ± 0.067	21 ± 1.9	68 ± 4	2.19 ± 0.104

Data are means ± standard deviation (SD) (n = 5–10).

## Data Availability

Data are contained within the article and Appendix A.

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
