# Peer review of "Investigation of the Nanoparticulation Method and Cell-Killing Effect following the Mitochondrial Delivery of Hydrophobic Porphyrin-Based Photosensitizers"

_ijms, 2024, doi:10.3390/ijms25084294_

Round 1

Reviewer 1 Report

Comments and Suggestions for Authors

The manuscript “Investigation of the nanoparticulation method and cell-killing effect following the mitochondrial delivery of hydrophobic porphyrin-based photosensitizers” describes the continuation of research of photodynamic therapy (PDT) with porphyrin-based photosensitizer rTPA encapsulated into MITO-Porter, a nanocarrier targeting mitochondria. In this study, in addition to negative, anionic rTPA, two analogues were prepared, neutral and positive (cationic), via reaction of rTPA with 2-aminoethanol and ethylenediamine, respectively, and they were encapsulated into MITO-Porter with either low, or high drug/lipid ratio. Although positive rTPA@MITO-Porter did not show better retention in the mitochondria than the others as the authors hypothesized, the obtained results can be valuable, and the topic is certainly very interesting.

The manuscript is well written, introduction with well-covered previous, relevant research and references, and the results are presented clearly and well discussed. I have only few questions and suggestions concerning experimental part.

It is not clear what is the light dose (i.e. duration of irradiation) that was used for in vitro PDT experiments (results shown in Fig 5). In Figure 4, for example, it says that irradiation was for 60 s. Furthermore, it is not clear if there was a ‘light-only’ control.

Could you please comment: Since carboxylate salt of rTPA is used as anionic photosensitizer, why not using also salt for cationic photosensitizer (e.g. obtained by alkylation or protonation of amino groups) for positive rTPA@MITO-Porter?

It would be useful to show in Supplementary information 1H NMR spectra of two new compounds, and purity of the synthesized compounds before encapsulation.

Reviewer 2 Report

Comments and Suggestions for Authors

This interesting manuscript describes the optimization of the nanoparticled photosensitizer rTPA/MITO-porter to enhance its tropism to mitocondria while retaining effective ROS-generating properties upon light irradiation. The study rationale is sustained by the authors’ previous studies on such compounds, the experiments have been properly designed and carefully performed and the results are convincing and well illustrated.

Major points

1. Section 2.5, evaluation of mitochondrial retention: a simple additional experiment to provide at least preliminary information on the mitochondrial retention of the 3 types of rTPA should be carried out, e.g. by washing the cells and looking at residual colocalization signal upon adequate time gaps in confocal microscopy. In my opinion, this experiment should not be matter for a future study but should rather be performed in this one: in fact, information on mitochondrial retention represents the main rationale for this study as well as a key point of novelty of the current study in respect with the previous ones already published by the same authors on these compounds.

2. Please do not use acronyms in the abstract. Similarly, please take care of spelling out all non-standard acronyms at their first mention (and avoid them if not often repeated). This comment also applies to the legends to the figures and to the supplementary file. Note that a general reader should be able to easily understand the experiments and find any needed information to repeat them, while jargon acronyms impede.

Minor points

3. Please explain how SOSG probe works: is fluorescence proportionally excited or quenched by singlet oxygen? The legend to figure 4 also needs further explanation: what does PL mean?

4. Section 2.4. intracellular uptake test: is it possible that NPD-DOPE-labeled nanoparticles behave differently than plain nanoparticles in terms of cell permeation capability? This point deserves a comment.

5. Section 2.5. mitochondrial localization: please report the incubation time of the confocal experiment shown in figure 7. Scattered misspellings/mistakes: lines 179-180, redundant doubled sentences; line 327, ‘followed by analysis’; line331, ‘observation using CLSM’; line 332, ‘the HeLa cells were seeded in 35-mm’.

Comments on the Quality of English Language

Adequate (minor misspellings)

Round 2

Reviewer 2 Report

Comments and Suggestions for Authors

Although I am still in the opinion that additional data on mitochondrial retention by the cells would have added substantial strenght to this study, I acknowledge that the close deadline for publication of the special issue to which this article is destined can be an acceptable reason for the authors’ denial to comply with my suggestion. The remaining points have been satisfactorily addressed.